# Multimodal Radiobioconjugates of Magnetic Nanoparticles Labeled with ^44^Sc and ^47^Sc for Theranostic Application

**DOI:** 10.3390/pharmaceutics15030850

**Published:** 2023-03-05

**Authors:** Perihan Ünak, Volkan Yasakçı, Elif Tutun, K. Buşra Karatay, Rafał Walczak, Kamil Wawrowicz, Kinga Żelechowska-Matysiak, Agnieszka Majkowska-Pilip, Aleksander Bilewicz

**Affiliations:** 1Department of Nuclear Applications, Institute of Nuclear Sciences, Ege University, Izmir 35100, Turkey; 2Centre of Radiochemistry and Nuclear Chemistry, Institute of Nuclear Chemistry and Technology, Dorodna 16 St., 03-195 Warsaw, Poland

**Keywords:** SPION, multimodal nanoparticles, PET diagnosis, MRI, ^44/47^Sc, PSMA-617, prostate cancer

## Abstract

This study was performed to synthesize multimodal radiopharmaceutical designed for the diagnosis and treatment of prostate cancer. To achieve this goal, superparamagnetic iron oxide (SPIO) nanoparticles were used as a platform for targeting molecule (PSMA-617) and for complexation of two scandium radionuclides, ^44^Sc for PET imaging and ^47^Sc for radionuclide therapy. TEM and XPS images showed that the Fe_3_O_4_ NPs have a uniform cubic shape and a size from 38 to 50 nm. The Fe_3_O_4_ core are surrounded by SiO_2_ and an organic layer. The saturation magnetization of the SPION core was 60 emu/g. However, coating the SPIONs with silica and polyglycerol reduces the magnetization significantly. The obtained bioconjugates were labeled with ^44^Sc and ^47^Sc, with a yield higher than 97%. The radiobioconjugate exhibited high affinity and cytotoxicity toward the human prostate cancer LNCaP (PSMA+) cell line, much higher than for PC-3 (PSMA-) cells. High cytotoxicity of the radiobioconjugate was confirmed by radiotoxicity studies on LNCaP 3D spheroids. In addition, the magnetic properties of the radiobioconjugate should allow for its use in guide drug delivery driven by magnetic field gradient.

## 1. Introduction

In nuclear medicine, nanoparticles (NPs) with magnetic properties can be used for imaging, diagnosis, treatment, and separation of biological materials [1]. Initial studies on patients with magnetically controlled drug targeting were reported by Lübbe et al. The authors stated that 0.5–0.8 T magnetic field intensity is sufficient to direct iron nanoparticles to tumors near the surface [2]. It is also possible to target drugs to diseased tissue by loading drug molecules on magnetic particles. Magnetic nanoparticles (MNPs) can also be labeled with radionuclides, which enables them to be used in radiopharmacy. MNPs may be advantageous in alpha radionuclide therapy to retain ^225^Ac and its daughter products at a target site. Cedrowska et al. reported that radiolabeled iron oxide nanoparticles with Ac-225 and modified with CEPA-Tmab were proposed for a combination of magnetic hyperthermia and radionuclide therapy [3]. In another publication ^223^Ra-doped BaFe nanoparticles were presented as candidates for multimodal drug combining localized magnetic hyperthermia with internal α-therapy [4]. Magnetic nanoparticles labeled also with β^−^ emitters (^188^Re, ^198^Au, ^90^Y, and ^131^I) and Auger electron emitters (^111^In and ^125^I) were also investigated, some of them in preclinical studies [5,6,7,8,9].

MNPs in medicine can be classified as therapeutic (hyperthermia and drug targeting) and diagnostic (NMR imaging) agents. Shape and size, biocompatibility, and stability of nanoparticles are parameters to be considered. Silica coating prevents magnetic nanoparticles from aggregating uncontrollably and oxidizing over time. TEOS (tetraethyl orthosilicate) is one of the most commonly used methods although there are different coating methods [10]. The silica layer stabilizes the MNP core by sustaining magnetic dipole interactions. In medical applications, MNPs can be coated with a biocompatible polymer to increase stability and bioavailability [11] or coated with antibody for biomarker immobilization [12]. Among others, MNPs modified with hyperbranched polyglycerol (HPG) have attracted much attention for years [1,2,3,4,5,6,7,8,9,10,11,13,14,15]. Having three-dimensional structures and numerous internal and external functional groups, these compounds can serve as remarkable hosts for metal complexes, enzymes, and biomaterials. HPG has many reactive functional groups that can be converted to other functional groups. In addition, HPG-modified MNPs have been used in various applications such as magnetic resonance imaging (MRI), drug delivery, and catalysis [16].

In our work, we designed a multifunctional agent combining PET and MRI imaging and radionuclide therapy using SPION nanoparticles as a platform [12]. A prostate-specific membrane antigen (PSMA) molecule is implemented for targeting, while a DOTA chelator incorporated into the PSMA-617 structure is used to complex two scandium radionuclides ^44^Sc for PET diagnosis and ^47^Sc for therapy.

The PSMA small molecule is an antagonist with a very high affinity to specific membrane antigens expressed in aggressive prostate cancer. Due to the high resistance of this tumor, it is recommended to use several therapeutic methods, e.g., chemo- and radiotherapy during treatment. The second therapeutic method planned for use is magnetic hyperthermia. The therapeutic effect obtained results from the impact of β-radiation emitted by ^47^Sc radiation, as well as from local hyperthermia—heating of cancer cells to a temperature above 42 °C, induced by fast oscillations of SPIONs in an external alternating magnetic field. Covalent binding of vector molecules (PSMA) for further radiolabeling in the final step of synthetic processes to the SPIONs provides a precise, targeted delivery of obtained radiobioconjugate only to the selected tumor cells overexpressing PSMA receptors, while the external alternating magnetic field causing an increase temperature multiplies the therapeutic effect.

It is well known that prostate cancer can spread to any part of the body, but metastatic sites are most commonly directed to the bones [17]. Nuclear medicine aims to treat bone metastases with [^153^Sm]Sm-EDTMP or [^177^Lu]Lu-EDTMP, while [^99m^Tc]Tc-MDP is mainly used in the imaging of bone metastases [18]. Phosphate derivatives such as EDTMP and DPAPA may be suitable targeting agents for theragnostics, where imaging and treatment with the same agent are used together. In this work, we added to HPG-modified MNPs two phosphonate derivatives, EDTMP or DPAPA, for targeting of prostate cancer and its metastases. With PSMA-617, we present a compound that, in addition to a PSMA inhibitor as a target vector, also contains a bisphosphonate that is established as a bone tracer and, thus, combines the advantages of PSMA targeting and bone targeting.

Additionally, due to their superparamagnetic behavior, radionuclide labeled SPIONs bioconjugates can be guided and retained exclusively in the tumor tissue with help of an external constant magnetic field. Their accumulation in this tissue can be followed by functional NMR and positron emission tomography (PET). Thus, we proposed a promising radionuclide therapy and imaging tool as an “all-in-one” approach.

## 2. Materials and Methods

### 2.1. Reagents and Instruments

Iron(III) acetylacetonate [Fe(acac)_3_; Fe(C_5_H_7_O_2_)_3_; 97%], decanoic acid [CH_3_(CH_2_)_8_COOH; 98%], benzyl ether [(C_6_H_5_CH_2_)_2_O; 98%], tetraethyl orthosilicate [TEOS; (C_2_H_5_O)_4_Si; for synthesis], potassium methylate (CH_3_OK; for synthesis), anhydrous methanol (CH_3_OH; 99.8%), glycidol (C_3_H_6_O_2_; 96%), ethylenediamine (NH_2_CH_2_CH_2_NH_2_; absolute, ≥99.5%), sodium cyanoborohydride (NaBH_3_CN; reagent grade, 95%), N-ethyl-N′-(3-dimethylaminopropyl)carbodiimide hydrochloride [E.D.C.; C_8_H_17_N_3_ hydrochloric acid (HCl) Bio extra], N-hydroxy succinimide (C_4_H_5_NO_3_; 98%), 3-(4,5-dimethylthiazol-2-yl)-2–5-diphenyltetrazolium bromide (MTT; 98%), dimethyl sulfoxide (DMSO), and trifluoroacetic acid [TFA; CF_3_COOH; high-performance liquid chromatography (HPLC), 99%] were obtained from Sigma-Aldrich (Taufkirchen, Germany). Acetonitrile (HPLC grade) was purchased from Carlo Erba Reagents (Barcelona, Spain). Dulbecco’s modified Eagle’s medium (DMEM), fetal bovine serum (FBS), penicillin/streptomycin solution, RPMI-1640, L-glutamine, trypsin-EDTA, nonessential amino acids, sodium pyruvate, and phosphate-buffered saline (PBS) were acquired from Biowest (Nuaillé, France). Muse^®^ Annexin V and Dead Cell Kit and Instrument Cleaning Fluid were purchased from Luminex (Northbrook, IL, USA). HCl (37%), phosphoric acid, and sodium hydroxide were supplied by Merck KGaA (Darmstadt, Germany). Deionized water was processed in a Milli-Q water purification system.

PC-3 [PSMA (−) human prostate derived from the metastatic part of the bone] and LNCaP [PSMA (+) prostate derived from the metastatic site left supraclavicular lymph node] cells were supplied by the American Type Culture Collection (ATCC; Manassas, VA, USA). These cells were obtained from the ATCC using the authors’ previous project resources for academic studies in the cell culture laboratory of the Ege University Institute of Nuclear Sciences.

The following reagents were also used: hydrochloric acid 35–38%, analytical pure, Chempur, Poland; ammonium acetate, analytical pure, Chempur, Poland; ammonia solution 25%, analytical pure, Chempur, Poland; human serum, Sigma-Aldrich, USA; Dulbecco’s PBS, Biological Industries, Israel; calcium carbonate 4.3% ^46^Ca, Isoflex, USA; calcium carbonate, 99.999%, Alfa Aesar, USA; syringe filter 0.2 μm, PTFE Whatman, Great Britain; cation-exchange resin Dowex 50WX4, mesh 100–200, H+, Fluka Analytical, USA.

The following materials were used: growing media—DMEM (PC-3) and RPMI 1640 (LNCaP); trypsin EDTA solution C; water, cell culture grade; phosphate-buffered saline (PBS); fetal calf serum from Biological Industries (Beth Haemek, Israel). For cytotoxicity evaluation, CellTiter96^®^ AQueous One Solution Reagent (MTS compound) from Promega (Mannheim, Germany) was used. LNCaP and PC-3 cells were obtained from the American Type Tissue Culture Collection (ATCC, Rockville, MD, USA) and cultured according to the ATCC protocol. For experimental applications, over 80% confluent cells were used. Blocking the PSMA receptors was accomplished with PSMA-617, obtained from Selleck Chemicals L.L.C., Houston, TX USA.

The following equipment was used: HPLC SPD-10AV ultraviolet–visible (UV–vis) and AD2 detector systems with an LC-10Atvp pump (Shimadzu, Kyoto, Japan), Inertsil ODS-3 C-18 4.6 × 250 mm HPLC 5 µm column (G.L. Sciences, Inc., Tokyo, Japan), SIL-20A HT automatic sampler (Shimadzu), Varioskan Flash multimode microplate reader (Thermo Fisher Scientific, Darmstadt, Germany), AR-2000 radioTLC (thin-layer radiochromatography) imaging scanner (Eckert & Ziegler, Berlin, Germany), Packard Tricorb-1200 liquid scintillation counter (Meriden, CT, USA), Malvern Zetasizer Nano ZS dynamic light scattering (D.L.S.; Malvern Panalytical, Malvern, U.K.), Millipore Muse Dead Cell Analyzer Flow Cytometry inverted microscope (Leica Microsystems, Wetzlar, Germany), and Spectrum Two I.R. spectrophotometer (attenuated total reflection; Perkin-Elmer, Boston, MA, USA). The Ege University Central Research Laboratory supplied the following analyses: X-ray photoelectron spectroscopy (XPS) analysis was conducted with the K-Alpha XPS System (Thermo Fisher Scientific, U.K.). The scanning probe microscopy (S.P.M.) image was taken using a Bruker Dimension Edge with ScanAsyst System (Billerica, MA, USA). Scanning electron microscopy (SEM) images were taken with a Thermo Scientific Apreo S device at the Ege University Central Research Laboratory and the SEM Zeiss EVO LS10 (Carl Zeiss Microscopy GmbH, Germany) at the Konya Selçuk University Advanced Research Center (Iltek). ^1^H-NMR, ^13^C-NMR, and ^31^P-NMR spectra of DPAPA were performed with the 400 MHz operating frequency liquid MERCURY plus-AS 400 model NMR spectrometer in the Nuclear Magnetic Resonance Laboratory of Ege University Faculty of Science. Transmission electron microscopy (TEM) measurements were made with a JEOL-2100 Multipurpose 200 kV TEM (Tokyo, Japan) at the Advanced Research and Application Center of Konya Selçuk University (Konya, Turkey). Vibrating sample magnetometer (V.S.M.) measurements were conducted at the Dokuz Eylül University Center for Fabrication and Application of Electronic Materials using Dexing Magnet VSM 550 devices, ICP-MS analyses were performed the Agilent Technologies 7800 Series device, C.A., United States in Izmir Katip Çelebi University Central Research Laboratory (MERLAB).

### 2.2. Cubic Fe_3_O_4_ (C-Fe_3_O_4_) NP Synthesis

Fe_3_O_4_ nanocubes were prepared according to the procedure described by Martinez-Boubeta et al. [19] involving heating a solution of Fe(acac)_3_, decanoic acid, and dibenzyl ether. This method is performed by the discriminant separation of nucleation and growth stages caused by the intermediate formation of the iron(III) decanoate complex, as discussed previously [19].

### 2.3. Silica and HPG Coating of C-Fe_3_O_4_ NPs and Ethylenediamine Coupling

Silica coating of C-Fe_3_O_4_ NPs was performed using the procedure described previously [20]. The HPG coating was applied according to the method proposed by Sadri et al. [21]. The synthesis path of ethylenediamine coupling with Fe_3_O_4_–SiO_2_–HPG in the literature was realized with modifications [22].

### 2.4. Synthesis of DPAPA and Conjugation with Fe_3_O_4_–SiO_2_–HPG–NH_2_

DPAPA was synthesized using a modified Mannich-type reaction according to the method given in the literature described for EDTMP [23] (Figure 1). Synthesized cubic MNPs were conjugated with DPAPA using the EDC/(N-(3-dimethyl aminopropyl)-N′-ethyl carbodiimide hydrochloride/N-hydroxy succinimide (NHS) conjugation method [20].

### 2.5. Production of ^44/47^Sc

***^44^Sc***: CaCO_3_ (88.87 mg, 99.999% purity) target material was pressed into a 6 mm disc, supported on graphite. The target was irradiated by 16 MeV proton with 14 µA current on GE-PET trace 840 cyclotron at Heavy Ions Laboratory, Warsaw University. The irradiation was carried out for 100 min.

***^47^Sc:***^47^Sc was produced in ^46^Ca(n,γ)^47^Ca→^47^Sc reaction in Nuclear Reactor—Maria in Świerk (Otwock, Poland). Then, 5 mg of CaCO_3_ (4.3% ^46^Ca) was irradiated for 120 h with 10^14^ n/cm^2^s thermal neutron flux. After irradiation, the target was cooled for 5 days to reach the maximum activity of ^47^Sc produced from ^47^Ca decay. Then, 5 days after the irradiation, 90 MBq of ^47^Sc was produced from ^47^Ca decay.

### 2.6. Separation Procedure

For the separation of ^44/47^Sc from the target material, a two-step method based on microfiltration [24] and cation exchange on Dowex resin was used [25]. First, the irradiated target was dissolved in 1 M HCl, and then the solution was alkalized with the ammonia solution (25%) to pH ~11. In these conditions, Sc compounds were trapped on 0.22 µm porous microfilters. The loss of radionuclides after this process was no higher than 10%. For the removal of Ca tracers, the microfilter was washed with 5 mL of deionized water. After that, scandium cations were removed from filters with 2 mL of 1 M HCl with an efficiency of 84.6% ± 0.75% and a 200 mg load of Dowex 50WX4 cation exchange resin. The bed was washed with 5 mL of water for removal of residual HCl with almost no loss of the activity. Elution of Sc from DOWEX resin was performed with 0.4 M ammonium acetate buffer, pH 4.5, 0.5 mL/min flow and the eluent was collected with 0.5 mL fractions.

### 2.7. Radiolabeling with ^44/47^Sc

After the separation process, ^44/47^Sc was in 0.4 M ammonium acetate buffer, pH 4.5, which is a suitable environment for labeling of DOTA chelator. Then, 1 mL of nanoparticles (41 mg of NP) were centrifuged 4500 rpm for 5 min. After that, supernatants were removed and replaced with 100 µL of 1 M ammonium acetate buffer. Next, 30 MBq of ^44^Sc or ^47^Sc in ammonium acetate buffer were added to EDTMP-PSMA NCs or DPAPA-PSMA NCs and incubated for 30 min at 95 °C. After incubation, the samples were centrifuged to remove the unbound ^44/47^Sc from the nanoparticles and their activities were measured. In every case, the efficiency of labeling was higher than 97%.

### 2.8. Stability Tests of ^44/47^Sc

Stability tests were performed for EDTMP PSMA NPs and EDTMP PSMA NPs labeled with ^44^Sc. After labeling, NPs were purified from nonattached scandium and divided onto two parts. Stability of bonding between Sc and NPs was checked in 500 µL of human serum (HS) or phosphate-buffered saline (PBS). Samples were incubated at 37 °C, and the stability was checked after 1, 2, 3, 4, and 24 h. After incubation at 37 °C nanoparticles were centrifuged, and the activity was measured.

### 2.9. In Vitro Studies

The PC-3 (PSMA−) and LNCaP (PSMA+) prostate cancer cell lines were used for cell culture studies. These cells were obtained from the American Type Culture Collection (ATCC) (USA) using our previous project resources for academic studies at Ege University Institute of Nuclear Sciences Cell culture laboratory. PC-3 was grown in medium consisting of Dulbecco’s modified Eagle’s medium (DMEM) and 10% fetal bovine serum (FBS). LNCaP was grown in medium consisting of Roswell Park Memorial Institute RPMI-1640 medium and 10% fetal bovine serum (FBS). Cells were incubated in 5% CO_2_ and 37 °C. The medium was changed every 2 days, and fresh medium was added. After the cells proliferated to cover 80% of the flasks, they were separated from the flask with 0.25% (W/V) trypsin-EDTA solution and planted in 96-well plates for cytotoxicity studies. Cells not used in the study were placed in media containing 5% DMSO, first frozen at −80 °C, and then stored in liquid nitrogen at below 198 °C, where they were stockpiled for further studies. All cell culture studies were performed in 6 replicates (n = 6).

#### 2.9.1. Cell Binding of ^44^Sc-Labeled Bioconjugates

***^44^Sc:*** Receptor binding affinity of synthesized bioconjugates was determined with LNCaP cells overexpressing PSMA receptors, as well as with PC-3 cells (PSMA-negative) used as control. Two days before the experiment, cells (8 × 10^5^ LNCaP and 5 × 10^5^ PC-3 cells, respectively) were seeded into six-well plates and incubated in 37 °C with 5% CO_2_ atmosphere. Subsequently, prior tested compounds were added, and the cells were washed once with PBS. Next, 1 mL of various concentrations (0.06–2.0 nM) of bioconjugates labeled with ^44^Sc (25–30 MBq) suspended in a growing medium were added and incubated for 2 h with slight shaking. Then, the medium was aspirated, and cells were rinsed with PBS twice to remove unbound fraction. In the last step, the cells (bound fraction) were lysed twice with 1 M NaOH, and all fractions were measured using a Wizard2 Detector Gamma Counter (Perkin Elmer, Waltham, MA, USA). For evaluation of nonspecific binding, PSMA receptors were blocked with 2000 M excess of nonconjugated PSMA. To calculate the specific binding, the difference between total and nonspecific binding was quantified. Presented results (mean with SD) contain the data from two individual experiments, wherein each sample was repeated twice.

#### 2.9.2. MTT Assay

Cytotoxicity tests were performed using the MTT (3-[4,5-dimethylthiazol-2-yl]-2,5 diphenyl tetrazolium bromide) method. The MTT test is based on the conversion of MTT into formazan crystals by living cells, which determines mitochondrial activity [18]. Because total mitochondrial activity correlates with the number of viable cells for most cell populations, this assay is widely used to measure the in vitro cytotoxic effects of drugs on cell lines or primary patient cells. Cells were prepared from cell suspensions at 5 × 10^4^ cells/mL per well of 96-well plates. Then, 100 µL of cell suspension was added to each well created, and a solution containing the sterile substance at six different (1, 3, 10, 30, 100, and 300 µg/mL) concentrations was added to the wells except the control. Cells and the reagent-free medium were used as a negative control. In the study, each parameter was studied with n = 5 repetitions. The plate with the cells was incubated at 37 °C in 5% CO_2_. At the 24th, 48th and 72nd hours of cell incubation (PC-3 or LNCaP), 10 µL of MTT solution was added to each well and incubated for 4 h. Next, after 4 h of incubation, instead of removing MTT, 100 µL of SDS (sodium dodecyl sulfate) was added and incubated at 37 °C for 24 h in a 5% CO_2_ environment. Cells were read using a spectrophotometer for the absorbance value of each well at a wavelength of 540 nm and a reference range of 690 nm.

#### 2.9.3. MTS Assay

Cytotoxicity studies with ^47^Sc were performed with MTS assay according to the previously reported protocol [5]. LNCaP (10 × 10^3^) and PC-3 (7.5 × 10^3^) cells were seeded into 96-well plates 48 h before treatment and incubated in 37 °C with 5% CO_2_-supplemented atmosphere. Subsequently, the medium was replaced with tested compounds labeled with ^47^Sc and suspended in a growing medium. Cells were incubated with nanoparticles for 48 h and 72 h. Following incubation, the medium was removed, and fresh medium was added. Lastly, 20 µL of CellTiter96^®^AQueous One Solution Reagent (Promega, MDN, USA) was added for 2 h incubation. Absorbance was measured at 490 nm to calculate the percentage of metabolically active cells. IC50 was calculated using GraphPad Prism v.8 Software (GraphPad Software, San Diego, CA, USA).

#### 2.9.4. 3D Cell Culture Studies

The “Hanging Drop Model” was used for 3D cell culture using with the LNCaP (PSMA+) and PC-3 (PSMA−) prostate cancer cell lines. LNCaP cells were amplified in a medium consisting of Roswell Park Memorial Institute RPMI-1640 medium and 10% fetal bovine serum (FBS), while PC-3 cells were derived from Dulbecco′s modified Eagle′s medium (DMEM) and 10% fetal bovine serum (FBS). Cells were incubated in 5% CO_2_ and 37 °C. The medium was changed every 2 days, and fresh medium was added. After the cells proliferated to cover 80% of the flasks, they were separated from the flask with 0.25% (*W*/*V*) trypsin-EDTA solution, and the confluent cell lines were first removed from the surface for 3D cell culture. After being removed and taken into the relevant medium, the total droplet volume was prepared as 50 µL and 50,000 cells, and 3D cell culture models were created in the “hanging drop plate”. The medium of the drops containing the cell population was changed every day, and spheroid formation was completed at the end of the 48th hour. After the spheroid was formed, fixation was performed to obtain a 3D cell image.

The formed spheroids were transferred into the chamber slide through the relevant medium. Then, it was washed with 200 µL of PBS, before adding 200 µL of paraformaldehyde and incubating at 4 °C for 30 min. At the end of the incubation, three washes were performed with 200 µL of PBS at 5 min incubation intervals. Then, 200 µL of Triton X-100 was added to the cell lines and incubated for 30 min at room temperature. After this stage, 100 µL (29 µg) of DPAPA and PSMA-DPAPA NCs, and 100 µL (4.5 µg) of EDTMP and PSMA-EDTMP NCs were added separately for each cell line chamber; the cells were incubated for 2 h in 5% CO_2_ and 37 °C. After 2 h, the applied substances were removed from the medium and washed three times with PBS. Lastly, 2 µL/mL DAPI was added, and the related cell images were taken using a fluorescent microscope with a fluorescent microscope (Olympus BX53) 10× green filter. At the same time, cell images were taken using “confocal microscopy 3D fluorescent imaging (Zeiss LSM880, Cambridge, UK)”.

#### 2.9.5. MRI of PC-3 and LNCaP Cells with Nanoconjugates

MR imaging of PC-3 and LNCaP cells incorporated with C-Fe_3_O_4_ nanoconjugates was conducted at T2 phase using a Siemens Verio 3T MRI Scanner (GmbH, Ettlingen, Germany) equipped with a 640 mT/m ID 115 mm gradient. Nanoparticles and nanoconjugates encoded with 1, 2, 3, and 4* were added to the LNCaP (PSMA+) and PC-3 (PSMA−) prostate cancer cell lines.

### 2.10. Statistical Analysis

Statistical analyses were performed using GraphPad Prism software version 8.0 for Windows (GraphPad Software, San Diego, CA, USA). To evaluate whether the collected numerical data are normally distributed to compare four unpaired groups, Kolmogorov–Smirnov normality tests were applied. The comparison of means between separate groups of numerical variables was performed using a one-way analysis of variance (ANOVA). Nonlinear regression analysis was performed with the GraphPad statistical program using the cytotoxicity (%) values. With this analysis, the IC50 (dose leading to death of 50% of the current cell population) values on the cell lines of all applied substances were determined.

## 3. Results and Discussion

### 3.1. Synthesis and Characterization of the Nanoparticles and Nanoconjugates

PSMA-DPAPA/EDTMP NC as a magnetite base nanoconjugate was synthesized to develop a multifunctional theranostic agent for imaging and therapy. First, cubic Fe_3_O_4_ covered with a layer of silica was synthesized. The silica coating was applied to increase the colloidal stability and biocompatibility of core Fe_3_O_4_ NPs. HPG, as a biocompatible, multifunctional, hyperbranched dendrimer polymer, was added to the structure to increase the biocompatibility and stability of the nanoconjugate [16]. The obtained samples were characterized by SEM and TEM microscopy. According to SEM and TEM images, C-Fe_3_O_4_ NPs (Figure 2a) were uniform homogeneous cubic crystal shaped with a particle size of about 38–50 nm similar to previously reported images of cubic NPs [19]. According to SEM images (Figure 2b) the shapes of C-Fe_3_O_4_-SiO_2_ NPs differ comparing to C-Fe_3_O_4_. This may depend on the result of the silica coating. In addition, SEM images of C-Fe_3_O_4_-SiO_2_-HPG samples show a polymeric structure related to HPG modification (Figure 2c).

Figure 2 shows SEM images of (a) C-Fe_3_O_4_ nanoparticles, (b) C-Fe_3_O_4_-SiO_2_, (c) C-Fe_3_O_4_-SiO_2_-HPG, and (d) DPAPA NCs. SEM images of C-Fe_3_O_4_ NPs showed that the cubic NPs agreed with previous reports (Figure 2a) [19]. High-density hydroxyl (–OH) groups on the outer surface of core Fe_3_O_4_ NPs represent the main source of reactive groups for the subsequent chemical surface. These hydroxyl groups on the surface of the nanoparticle core are combined with the silicic acid which are formed directly from sodium silicate, which polymerizes with the decrease in pH during the silication step. As a result, the resulting silanol groups (Si–OH) are condensed into covalent siloxane bonds (Si–O–Si), leading to the formation of the silica coating layer surrounding the particle core [26]. Moreover, silicated C-Fe_3_O_4_ nanoparticles do not aggregate easily as they exhibit higher stability compared to uncoated nanospheres. Therefore, the surface of C-Fe_3_O_4_-SiO_2_ NPs differs from that of C-Fe_3_O_4_ NPs according to SEM images (Figure 2b). SEM images of C-Fe_3_O_4_-SiO_2_-HPG also show the polymeric structure of HPG on the surface (Figure 2c) since HPG is a glycerol-derived polymer with a large molecular structure. HPG-coated NPs had wrinkled and folded structures with polymers covering the NP surfaces in the SEM image. Similar SEM images were obtained in studies with HPG [27]. The DPAPA NC SEM image (Figure 2d) shows that DPAPA binds to C-Fe_3_O_4_-SiO_2_-HPG, forming a less porous and more planar morphology. However, in closer images, the porous structure formed on the polymer surface could be seen due to the structural properties of DPAPA.

Figure 3 shows the TEM images of C-Fe_3_O_4_ (A), C-Fe_3_O_4_-SiO_2_ (B), and C-Fe_3_O_4_-SiO_2_-HPG (C), which are compatible with cubic Fe_3_O_4_ images in the literature [19]. Dark areas or black spots seen in TEM images belong to the Fe_3_O_4_ core, whereas the colorless parts belong to silica and HPG polymer due to their lower electron density. Iron gives a more intense appearance compared to other phases. In addition, Fe_3_O_4_ NPs are uniformly cubic-shaped with a size from 38 to 50 nm. TEM images showed that the Fe_3_O_4_ core is surrounded by SiO_2_ and the organic phase.

The hydrodynamic diameters, zeta potentials and PDI values of the nanoconjugates are given in Table 1. The hydrodynamic sizes of C-Fe_3_O_4_ NPs were found to be 116 nm, in accordance with other reports [28], and they increased when the nanoparticles were covered with layers of SiO_2_, HPG, and DPAPA. Hydrodynamic sizes involving the solvent molecules around the nanoparticles are, therefore, usually measured larger than the dimensions measured by SEM and TEM images. The size distribution of the samples is expressed by the PDI value, where the PDI value and the size homogenization are inversely proportional [29]. The data presented in Table 1 are consistent with the literature [30], indicating a homogenous population of nanoconjugates.

Zeta potentials (ζ) provide information about the charge distribution on the surface of the nanoconjugates. In general, absolute zeta potential values above 30 mV provide good stability and about 20 mV provide only short-term stability, which is generally considered as a threshold value for the electrostatic stabilization; values in the range −5 mV to +5 mV indicate fast aggregation [31,32]. We performed zeta potential measurements at three various pH (pH = 2, pH = 7, and pH = 12), where, at pH 2 and 12 the hydrodynamic diameter of the NPs did not show any significant changes. According to the literature data, zeta potential values become increasingly negative as the pH value increases [33,34,35]. At physiological pH (pH = 7.0), the zeta potential value of the DPAPA NC was −24.2 ± 0.30 mV, indicating threshold colloidal stability. At pH = 2, colloidal stability was lost, and zeta potential values of nanoconjugates were +3.20 ± 0.40 mV and +5.40 ± 0.50 mV for C-Fe_3_O_4_-SiO_2_-HPG and DPAPA NC, respectively. In addition, the C-Fe_3_O_4_-SiO_2_-HPG and DPAPA NCs had a zeta potential of −29.10 ± 4.10 and −38.80 ± 3.40 mV at pH = 12, which can be attributed to the –OH groups of HPG molecule. The obtained zeta potential values as a function of pH confirm the presence of amine-decorated HPG on the MNP surface. For aminated HPG-coated MNPs, the particles had higher potential because the introduction of aminated HPG changed the interfacial properties of the particles in the solution. The long molecule chains increased the “water solubility” of particles and protected the particles from congregating [36].

The obtained samples were also characterized by FTIR (Figure 4). The peak observed at wave number 550 cm^−1^ for C-Fe_3_O_4_ was related to Fe–O bond [37]. The peak at 1071 cm^−1^ belongs to the Si–O stretch band. FTIR spectra of C-Fe_3_O_4_-SiO_2_-HPG and C-Fe_3_O_4_-SiO_2_-HPG-NH_2_ showed Fe–O bonds in the 530 cm^−1^ band. Si–O bonds at 1023 cm^−1^, N–H stretch at 1455 cm^−1^, C=C tension at 1635 cm^−1^, C–H tension at 2874 cm^−1^, and the –OH peak at 3338 cm^−1^ confirm the HPG molecular structure [38]. P–O and P=O stresses were seen at 975 cm^−1^ and 1123 cm^−1^ in the FTIR spectra of C-Fe_3_O_4_-SiO_2_-HPG-NH_2_ and DPAPA conjugated NPs. The N–H stretch at 1458 cm^−1^ and C=O stretch at 1628 cm^−1^ of–COOH groups were observed, which is consistent with the literature [39]. Thus, FTIR studies confirmed the coating of magnetite nanoparticles with a layer of silica, HPG polymers, and DPAPA molecules.

The magnetic properties of nanoparticles and bioconjugates were measured by an applied magnetic field between −3000 and +3000 Gauss. The saturation magnetization of the SPION core was 60 emu/g; however, the coating of the SPIONs with silica and polyglycerol reduced the magnetization significantly. The saturation magnetizations were 25 and 0.1 emu/g for silica-coated magnetite nanospheres and HPG-coated nanoconjugates, respectively. NPs exhibited ferrimagnetic characteristics with a small coercivity value [28,33,40]. A similar value 25 emu/g was reported previously [41] for silica-coated magnetite NPs. Xu et al. pointed out that coating with nonmagnetic materials affected the magnitude of magnetization of the coated magnetic materials due to the quenching of surface moments [42]. Because, in our studies, magnetite NPs were coated with nonmagnetic silica and HPG polymer, a similar mechanism could be considered to decrease saturation magnetization after silica and HPG coating [28,32]. Unfortunately, the low values of saturation magnetization do not allow the use of synthesized bioconjugates in magnetic hyperthermia therapy.

### 3.2. ^44^Sc Radiolabeling of Nanoconjugates

The scandium radiolabeling yield was measured using a cyclotron producing ^44^Sc. First, 17.76 MBq ^44^Sc for EDTMP-PSMA and 17.76 MBq ^44^Sc for DPAPA-PSMA were added to NP’s and incubated for 30 min at 95 °C. After incubation, the samples were centrifuged to remove the unbound ^44^Sc from the nanoparticles, and their activities were measured. The radiolabeling efficiency for EDTMP-PSMA was found to be almost 98%. For DPAPA-PSMA, the radiolabeling efficiency was found to be almost 97%.

### 3.3. Stability of the Obtained Radiobioconjugates

For stability studies, human serum (HS) and PBS solutions was used. The samples labeled with ^44^Sc were incubated at 37 °C. After 1, 2, 3, 4, and 20 h of incubation, samples were measured and centrifuged, and then the supernatant was removed from NPs. NPs without supernatant were measured once more, and the stability of the connection between NP and Sc was determined. As expected, due to the strong complexation of ^44^Sc by the DOTA ligand, the stability of the radiobioconjugates was high (Table 2).

As expected, due to the high thermodynamic and kinetic stability of the DOTA complexes, the stability of the radiobioconjugates was also high.

### 3.4. Cell Studies

#### 3.4.1. Cell Binding of ^44^Sc-Labeled Bioconjugates

As presented in Figure 5, both tested compounds bound specifically to PSMA receptors. The significant (*p* < 0.05) decrease in bound fraction confirmed this during the receptor blocking with excess PSMA. The EDTMP- and DPAPA-based conjugates showed a similar percentage of total bound fraction, calculated as 6.6% (EDTMP) and 5.1% (DPAPA), while the specifically bound fraction was 4.6% and 3.1%, respectively. No binding was observed for PSMA-negative PC-3 cells, which directly shows that synthesized conjugates were successfully conjugated with PSMA, and their biological activity was maintained. PSMA-617-DPAPA/EDTMP NCs are PSMA receptor-specific. Loveless et al. showed that PSMA receptors on LNCaP cells were specifically targeted by using [^44/47^Sc]-PSMA-617 at a molar activity of 10 MBq/nmol [43].

#### 3.4.2. MTT Assay

The percentage viabilities decreased with time in PC-3 and LNCaP cell lines (Figure 6). PSMA-617 conjugated nanoconjugates had a less toxic effect on cells. The affinities of PSMA and PSMA conjugated nanoparticles to LNCaP cells were relatively higher since LNCaP cells are PSMA+ prostate cancer cells [44].

#### 3.4.3. MTS Assay

Neither EDTMP nor DPAPA conjugates affected the mitochondrial activity of LNCaP and PC-3 cells (Figure 7). This agrees with our expectations, because PSMA function is limited to effective targeting without any therapeutic demands due to its low concentration in bioconjugates. We chose nontoxic nanoconjugates concentration, according to the data shown in Figure 5 (1 ng/mL). A significant decrease in survival fraction was found after 48 and 72 h incubation of ^47^Sc labeled radiobioconjugates in LNCaP cells (Figure 7). After 48 h, both EDTMP and DPAPA induced a ~40% decrease in metabolically active cells regardless of the activity concentration (*p* ≤ 0.01). Subsequently, we found dose-dependent cytotoxicity progression after 72 h. The EDTMP-based radiobioconjugate efficacy was variable from 64.14% ± 5.2% (*p* ≤ 0.001) survived fraction (1.25 MBq/mL) to 35.75% ± 3.2% (20 MBq/mL; *p* ≤ 0.0001). DPAPA-based radiobioconjugates were slightly more cytotoxic, and 31.22% ± 1.9% of cells remained unaffected after 72 h incubation with 20 MBq/mL of ^47^Sc (*p* ≤ 0.0001). The calculated half maximal inhibitory concentration (IC50) also showed that DPAPA-based conjugates were more effective (IC50 = 5.3 MBq/mL) when compared to EDTMP (IC50 = 7.1 MBq/mL). Any impact of nonradioactive and radioactive conjugates was found for PC-3 cell line, additionally proving specific anticancer activity only against PSMA(+) cells.

#### 3.4.4. Three-Dimensional (3D) Cell Culture Studies

According to data obtained during MTS assay (Figure 7) we found a slight advantage of DPAPA-based NC over EDTMP-based NC. Taking into account the stronger effect (lower IC_50_ and lower cell viability after 72 h), we decided to investigate DPAPA-based NC against 3D cell cultures. In terms of fluorescence spectra, DPAPA and DPAPA NC nanoconjugates presented fluorescence at 290 nm excitation and 420 nm emission wavelengths. Taking advantage of their green fluorescence properties, these nanoconjugates were applied to the prepared PC-3 and LNCaP 3D spheroids, and fluorescence images were obtained (Figure 8).

Confocal microscopic images of PSMA-617-DPAPA NC and DPAPA NC applied to the PC-3 cells are shown in Figure 9 and Figure 10. In these images, PSMA-617-DPAPA NC and DPAPA NC are green-colored because of the fluorescence property of DPAPA, while DAPI-stained cell nuclei are seen as blue in the cell nucleus. Images are given with an overlapping blue and red filter.

Confocal microscope images were acquired at 500 nm low-resolution and suggested a maximum cell “thickness” of 9.5 µm. which was in excellent quantitative agreement with S.P.M. measurements in the cell nucleus region. These images also showed that the nanoconjugates were homogeneously distributed in the cell cytoplasm and that DPAPA retained its fluorescence (Figure 9 and Figure 10). DPAPA (similarly to EDTMP) shows the fluorescence properties, and a report with lanthanide complexes showed that it can be used in cell imaging by taking advantage of this property [45].

#### 3.4.5. MRI

MR contrast measurement revealed that the iron contrast could be seen in cell media, although limits were much more significant compared to the PET and SPECT methods (Figure 11). Both SPECT and PET have contrast measurement limits in the picomolar range, while MRI and CT have contrast measurement limits at much higher nmol concentrations [46]. The superior spatial resolution of PET (4–5 mm) makes it more attractive than SPECT (10–15 mm). However, the spatial and temporal resolution of both methods is significantly less than that achieved with MRI or CT. The high sensitivity of nuclear methods combined with the favorable resolution of CT and MRI is the driving force behind hybrid imaging systems such as PET/CT and PET/MRI now available.

The MRI imaging potential of Fe_3_O_4_ nanoconjugates in vivo is essential in terms of adding PET/MRI imaging to the theranostic property of these nanoparticles. In our previous studies, MRI images were taken of prostate tumor-bearing mice given FDG-linked Fe_3_O_4_ nanoparticles (FDG-MNP), and FDG-MNPs were concentrated in the prostate tumor. At the same time, relatively small amounts were found in organs of other tissues, particularly the spleen and liver, and FDG-MNP concentrations decreased over time in all tissues [47]. In vivo animal MRI images containing Fe_3_O_4_ and showing the Fe contrast of different conjugate nanoparticles are available in the literature [48,49].

Metastases of prostate cancer cells usually move to the bones. The pelvis and spinal bones are some of the most common areas where prostate cancer spreads. However, radionuclide-labeled phosphonate derivatives are effective for imaging bone metastases and in radionuclide bone pain therapy. While [^99m^Tc]Tc-diphosphosphonate imaging is used for imaging, ^153^Sm- or ^177^Lu-radiolabeled EDTMP was applied for radionuclide bone pain therapy. Combining both imaging and treatment with the same agent may be advantageous regarding theranostic potential. Therefore, in this study, a phosphate derivative EDTMP conjugate was added to the nanoconjugate enabling imaging of bone metastases with MR and PET techniques. Another advantage of this approach is to increase the hydrophilicity and solubility of the nanoconjugate.

## 4. Conclusions

In our work, we proposed a new solution that involves the use of multimodal superparamagnetic iron oxide-based nanoparticles (SPIONs), labeled with two scandium radionuclides, ^44^Sc and ^47^Sc, that allow for visualization of the cancer tissue and simultaneous direct irradiation of the tumor. The synthesized nanoparticles were conjugated with biologically active PSMA-617 molecules directing to and recognizing the targeted tumor tissue. The proposed multimodal SPIONs are advantageous because they can provide successful diagnosis and therapy even for chemo- and radioresistant tumor cells.

Prostate cancer bone metastases are common at the advanced stage of disease. This issue deserves particular attention due to its huge impact on patient management and the recent introduction of many new therapeutic options. Imaging of bone metastases is essential to localize lesions, determine their size and number, and examine features and changes during treatment. Therefore, in the present study, the phosphonates EDTMP and DPAPA were added to the nanoconjugate, which enabled MR and PET imaging of bone metastases and palliative therapy with β^−^ radiation emitted by ^47^Sc.

We think that a practical application of our proposal is a stable magnetic PSMA radiobioconjugate labeled with ^44^Sc and ^47^Sc, which according to the in vivo studies, can be used for aggressive prostate cancer. We also expect that our results can be helpful in designing other multimodal magnetic radiopharmaceuticals applied in simultaneous PET and NMR diagnosis and radionuclide therapy.

## Figures and Tables

**Figure 1 pharmaceutics-15-00850-f001:**
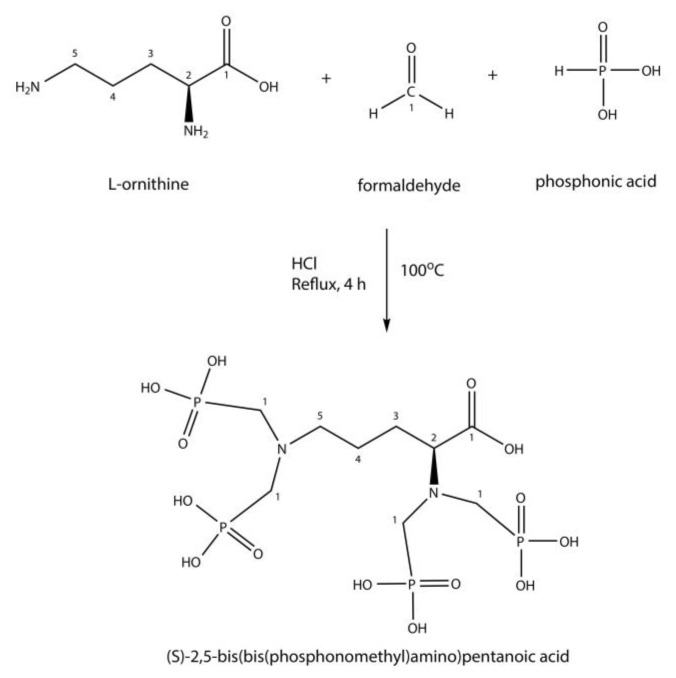
DPAPA synthesis reactions (Mannich-type reaction).

**Figure 2 pharmaceutics-15-00850-f002:**
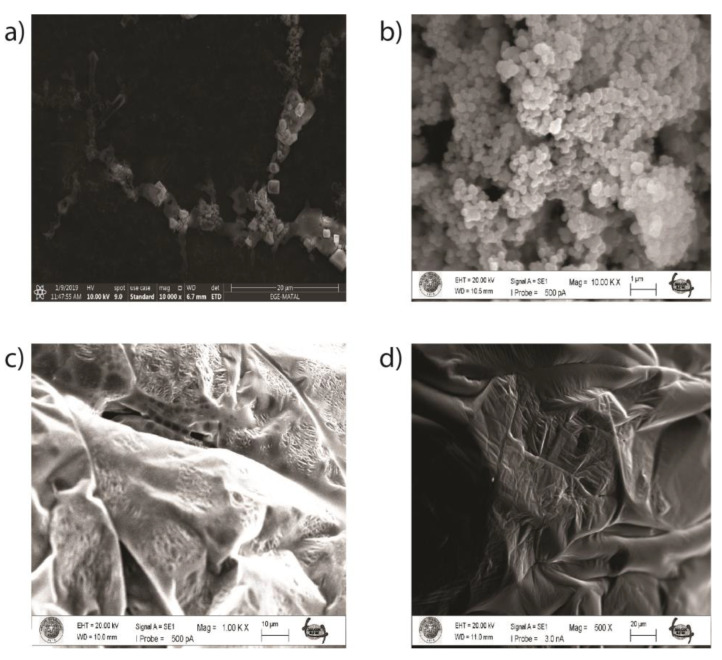
SEM image of obtained nanoparticles and nanoconjugates: (**a**) C-Fe_3_O_4_ SEM image, (**b**) C-Fe_3_O_4_-SiO_2_ SEM image, (**c**) C-Fe_3_O_4_-SiO_2_-HPG SEM image, and (**d**) DPAPA NC SEM image.

**Figure 3 pharmaceutics-15-00850-f003:**
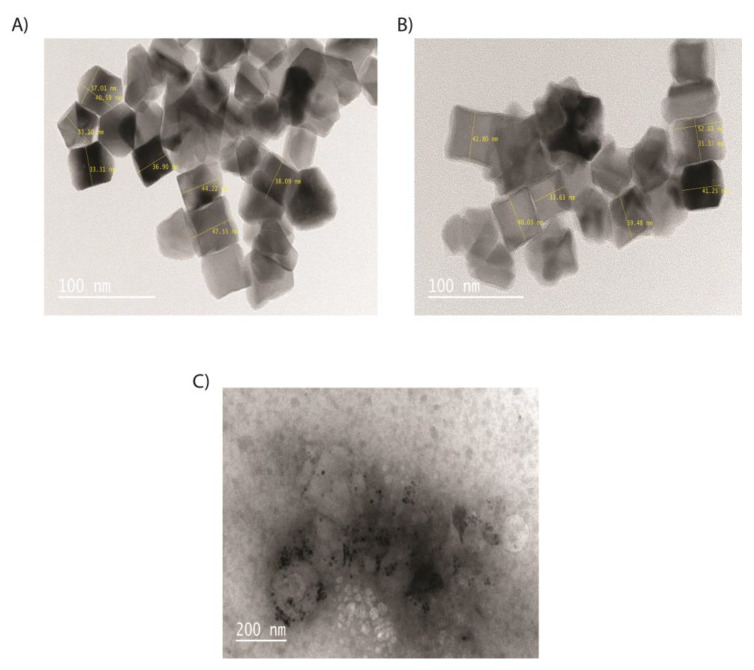
TEM images of NPs and nanoconjugates: (**A**) C-Fe_3_O_4_ NPs, (**B**) cubic silica-coated Fe_3_O_4_ NPs, and (**C**) cubic HPG-modified and silica-coated Fe_3_O_4_ NPs.

**Figure 4 pharmaceutics-15-00850-f004:**
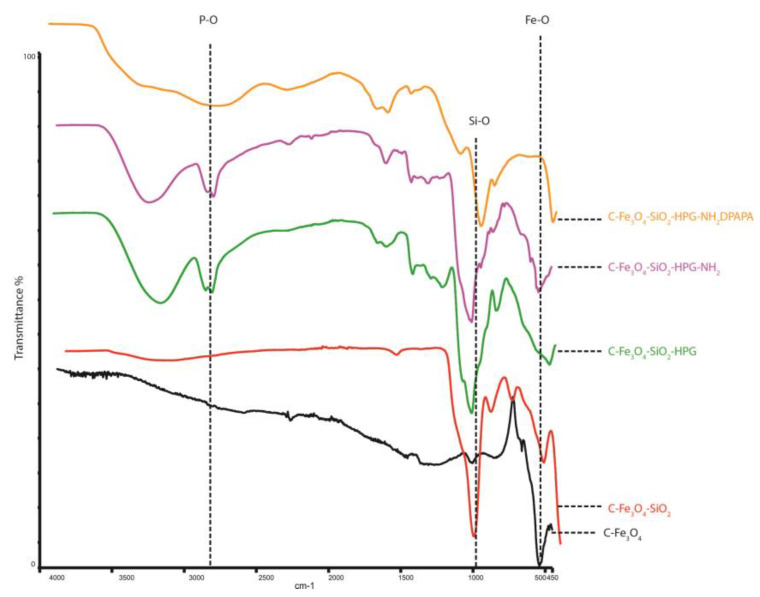
FTIR spectra of NPs and nanoconjugates.

**Figure 5 pharmaceutics-15-00850-f005:**
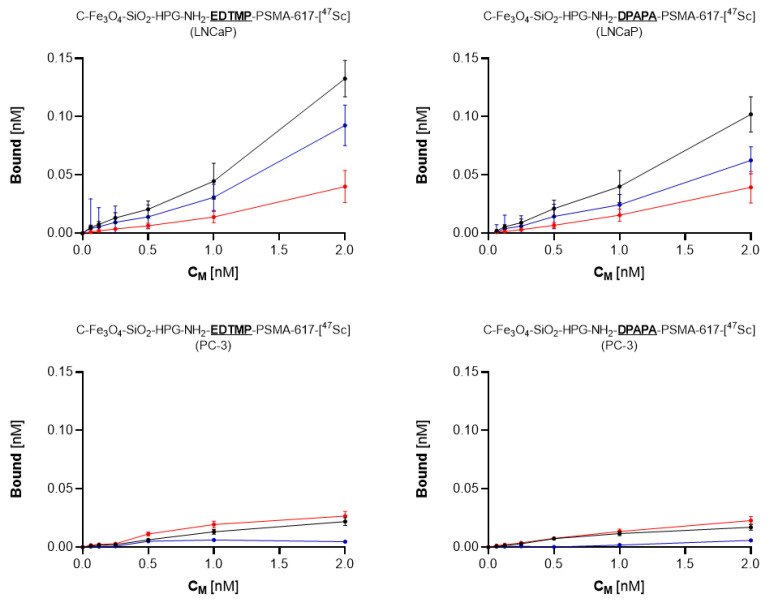
Receptor binding affinity studies of ^44^Sc labeled EDTMP- and DPAPA-based radiobioconjugates. Upper graphs show data for LNCaP cells; bottom graphs show data for PC-3 cells.

**Figure 6 pharmaceutics-15-00850-f006:**
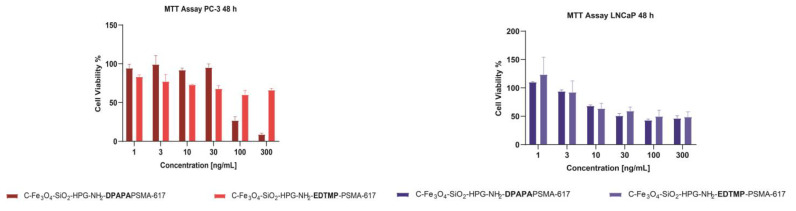
Cell viability of PC-3 and LNCaP cells treated with nanoconjugates after 48 h of incubation.

**Figure 7 pharmaceutics-15-00850-f007:**
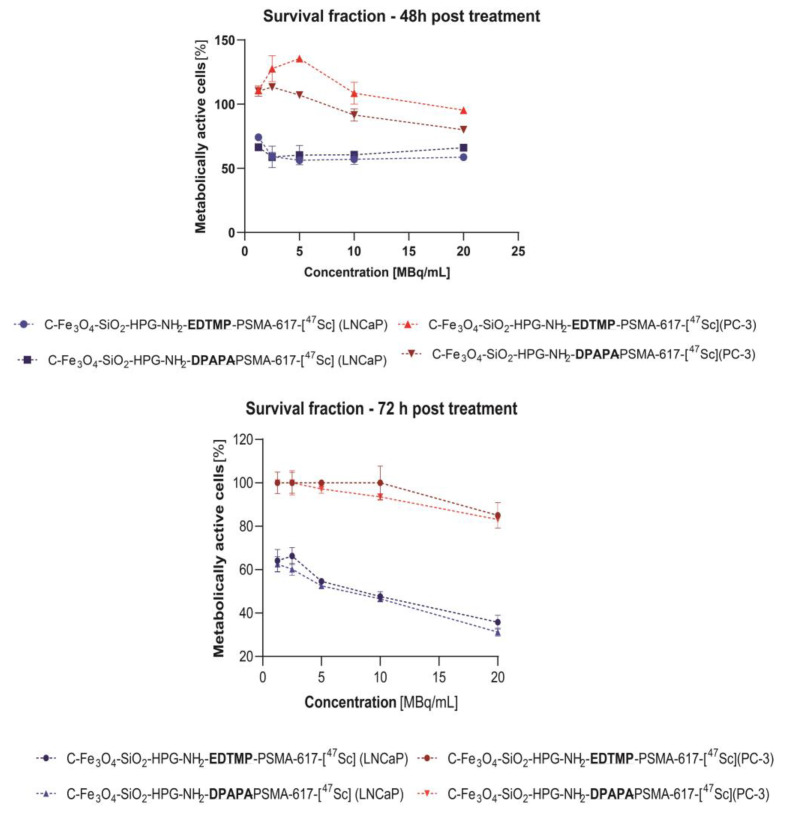
Cell viability of LNCaP and PC-3 cells treated with radiobioconjugates.

**Figure 8 pharmaceutics-15-00850-f008:**
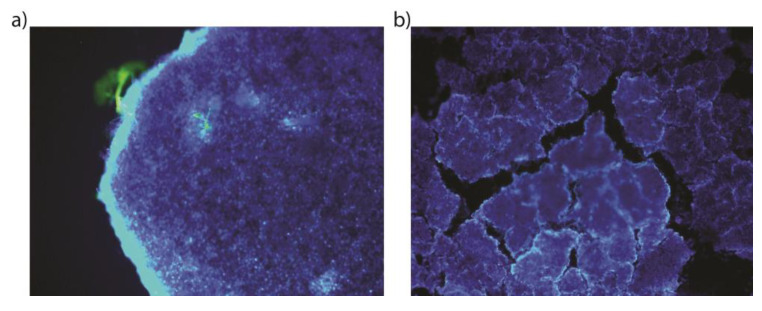
Three-dimensional fluorescence microscope images of (**a**) PC-3 cells (10× objective magnification, green filter) and (**b**) LNCaP cells (10× objective magnification, blue filter). The image of the nanoconjugates applied to the PC-3 cell line in the green filter can be seen. While cell nuclei are seen with DAPI, the intensely glowing region on the cell wall is thought to be the fluorescence feature from our nanoconjugates.

**Figure 9 pharmaceutics-15-00850-f009:**
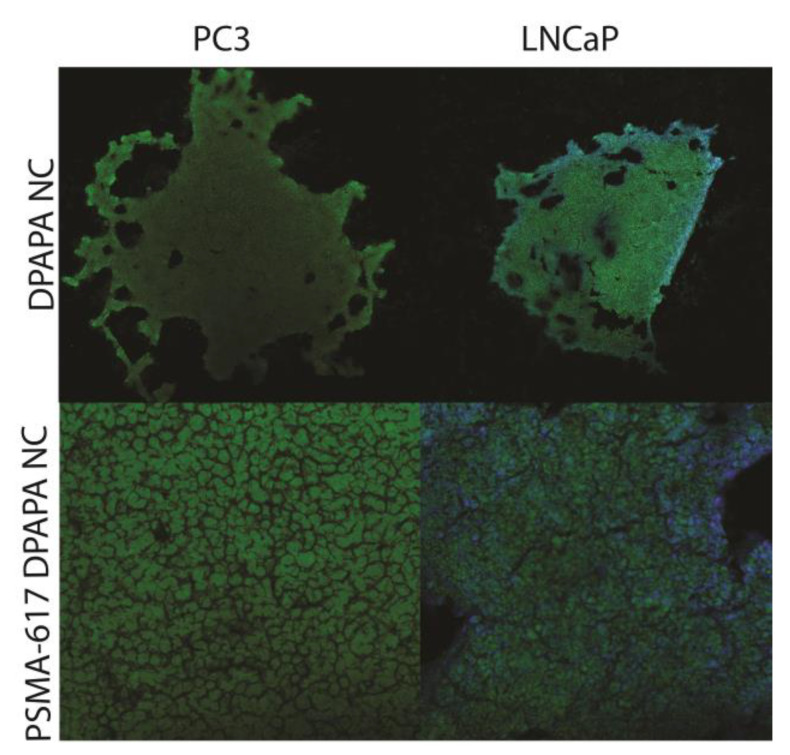
Images of C-Fe_3_O_4_-SiO_2_-HPG-NH_2_-DPAPA and C-Fe_3_O_4_-SiO_2_-HPG-NH_2_-DPAPA-PSMA-617 applied to PC-3 and LNCaP three-dimensional cell lines (10× objective magnification taken with blue and red filter).

**Figure 10 pharmaceutics-15-00850-f010:**
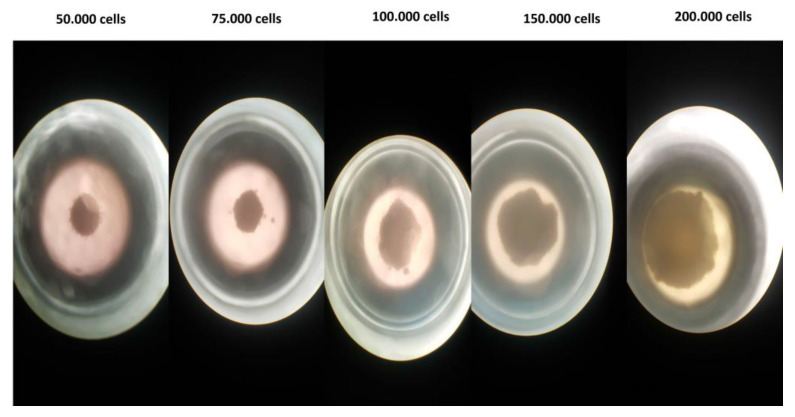
Spheroid images of 3D PC-3 cells (LEICA-DFC280 inverter microscope, 100×).

**Figure 11 pharmaceutics-15-00850-f011:**
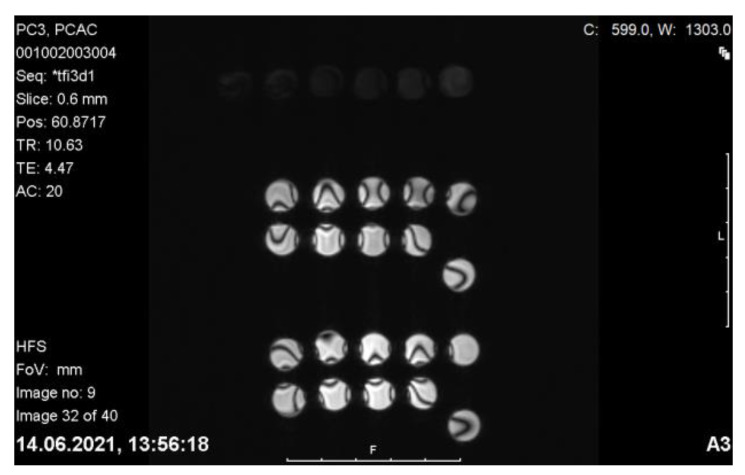
Relaxivity (R2): 0.421/3 T nanoconjugates applied on C-Fe_3_O_4_ and PC-3 LNCaP) (n = 6).

**Table 1 pharmaceutics-15-00850-t001:** Hydrodynamic sizes and zeta potentials of C-Fe_3_O_4_ NPs and nanoconjugates (n = 6).

Nanoconjugate	Hydrodynamic Size (d.nm) (n = 3)	Zeta Potential (mV) (n = 3) (pH = 7)	PDI
C-Fe_3_O_4_	116 ± 7.90	−18.6 ± 0.60	0.13
C-Fe_3_O_4_-SiO_2_	122 ± 0.20	−21.5 ± 0.01	0.22
C-Fe_3_O_4_-SiO_2_-HPG	145.8 ± 3.50	−18.5 ± 0.20	0.15
DPAPA NC	221.9 ± 16.00	−24.2 ± 0.30	0.06

**Table 2 pharmaceutics-15-00850-t002:** Stability of the ^44^Sc labeled C-Fe_3_O_4_ NC (n = 6).

	1 h	2 h	3 h	4 h	24 h
**EDTMP (HS)**	93.0	89.7	89.6	90.0	91.7
**EDTMP (PBS)**	98.9	97.4	97.3	98.3	96.0
**DPAPA (HS)**	95.3	94.3	92.1	92.2	84.0
**DPAPA (PBS)**	98.4	96.8	97.4	97.4	92.3

## Data Availability

Data are available on request.

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
