# Peer review of "Multimodal Radiobioconjugates of Magnetic Nanoparticles Labeled with 44Sc and 47Sc for Theranostic Application"

_pharmaceutics, 2023, doi:10.3390/pharmaceutics15030850_

Round 1

Reviewer 1 Report

This manuscript refers to the preparation, characterization and in vitro evaluation of 44/47Sc-labelled PSMA-targeted magnetic NPs conjugate. The quality of the manuscript is high. However, I have several comments and suggestions: 

 1. Introduction - page 2, line 48, I suggest to add few sentences about the labelling and use of alpha emitters labelled magnetic NPs, e.g. with 223Ra and 225Ac. 

Page 3, line 126, please correct "46Ca" 

Page 5, cell studies - Please always specify the number of experiment replicates, (e.g. n = 5). Please add this information also to Figure captions.

Page 8 - EM images. Please comment on the specific shape of coated NPs. c) and d). Is that only because of different magnification? 

Page 9, Table 1 - Please comment on the measured Z-potential values. It is generally accepted that values bellow -30 mV indicate good dispersions stability. did you observed any significant aggregation during the experiments / stability studies? Particularly when once centrifuged, it is quite complicated to redisperse such NPs.

Page 10, table 2.  Please specify the number of replicates performed or alternatively state single runs. 

Pages 10 and 11, Figures 4, 5, 6 - Please mark at the figures statistically significant differences between the data series. E.g. an asterix or other symbols. 

Author Response

  1. Introduction - page 2, line 48, I suggest to add few sentencesabout the labelling and use of alpha emitters labelled magneticNPs, e.g. with 223Ra and 225

We added few sentences about labelled magnetic NPs with 225Ac and 223Ra.

  1. Page 3, line 126, please correct "46Ca"

done

  1. Page 5, cell studies - Please always specify the number ofexperiment replicates, (e.g. n = 5). Please add this informationalso to Figure captions.

We added the number of experiment replicates.

  1. Page 8 - EM images. Please comment on the specific shape of coated NPs. c) and d). Is that only because of different magnification?

We added information to the manuscript. You are right, it is due to different magnification of the SEM images. Figure 2A, 2C and 2D have a magnification of 20 µm, while Figure 1B has a magnification of 1 µm.

  1. Page 9, Table 1 - Please comment on the measured Z-potential values. It is generally accepted that values bellow -30 mV indicate good dispersions stability. did you observed any significant aggregation during the experiments / stability studies? Particularly when once centrifuged, it is quite complicated to redisperse such NPs.

We added Z-potential values to the manuscript. We performed zeta potential measurements at three various pH (pH=2, pH=7 and pH=12). We had not observe any significant aggregation during the experiments and our nanoparticles were stable up to 24 hours (Table 2).

  1. Page 10, table 2. Please specify the number of replicates performed or alternatively state single runs.

Thank you for your comment. We added the number of replicates.

  1. Pages 10 and 11, Figures 4, 5, 6 - Please mark at the figures statistically significant differences between the data series. E.g. an asterix or other symbols.

We inserted into the manuscript text p-values to show statistical significance of obtained results discussed in the relevant section.

Reviewer 2 Report

The manuscript “Multimodal radiobioconjugates of magnetic nanoparticles labeled with 44Sc and 47Sc for theranostic application” presents an idea of using superparamagnetic iron oxide-based nanoparticles (SPIONs) labeled with two scandium radionuclides, 44Sc and 47Sc, for visualization of cancer tissue, simultaneous direct irradiation, and local magnetic hyperthermia of the tumor, although the magnetization results do not offer this possibility. Consequently, to avoid confusion to the reader, the hyperthermia part should be omitted in the approach.

It is striking that the authors measure the diameter in a cube. This is a misinterpretation of basic geometry. The diameter is the line segment that passes through the center and joins two opposite points of a circle. In 3D (sphere) it is defined as the segment that passes through the center and has its ends on the surface of the sphere. The authors should clarify this point. "...the Fe3O4 NPs have a uniform cubic shape and a diameter from 38 to 50 nm"

There is too much combination of ideas that make the reader lose understanding. The authors should improve the writing of the manuscript. For example, in the first paragraph of the introduction the authors present several ideas that somehow try to link them, but this reviewer finds the approach confusing.

It is preferable that the authors cite the original papers when they indicate that "micro and nanoparticles labeled 188Re, 198Au, 111In, 90Y, 125I and 131I have been prepared and used [3]." However, reference [3] is a paper where only 198Au is used and the others are listed in the introduction "Currently, many radionuclides are under intensive investigation for the therapeutic applications, particularly the Auger emitters..." this reviewer finds it bad practice to self-cite without clear justification.

The authors indicate that the coating occurs before synthesis of the MNPs "MNPs can be coated with a biocompatible polymer before or after synthesis..." I do not find this statement meaningful.

This reviewer does not find it valid to use a reference from almost 10 years ago to indicate that HPG has attracted attention in recent years.

The authors indicate that "Shape and size, biocompatibility, biodegradability, and stability of nanoparticles are parameters to be considered." Were all of these parameters considered in their work, i.e., are there results for all of these?

There are some statements without citing. As an example "The silica layer stabilizes the MNP core by sustaining magnetic dipole interactions." The authors should reinforce the ideas raised in the introduction by providing more references. For example, they could include https://doi.org/10.1016/B978-0-12-824364-0.00019-8 when talking about polymer-coated magnetic nanoparticles, they could also include https://doi.org/10.1155/2014/978284 to reinforce the approach of biomarker immobilization on silica-modified SPION surface.

The Objective of the work is clearly stated, however, it should not be written in the future.

The final product is too complex, perhaps difficult to replicate even for the same research group, it is cubic iron oxide nanoparticles, coated with silica, then another component is added, HPG, later all this is conjugated with DPAPA, and even more with PSMA. Is it so? None of the results show evidence of the formation of the proposed material, and the interaction between the materials is not discussed. There is no evidence or demonstration that the nanoparticles have been modified. The authors describe a number of characterizations, such as FTIR and magnetization measurement without providing graphs. Why are the graphs not presented?

Figures 2C and 2D do not show any nanostructure, at least it cannot be seen at that SEM resolution.

The authors indicate that "Because in our studies magnetite NPs were coated with nonmagnetic silica and HPG polymer, a similar mechanism could be considered to decrease saturation magnetization after silica and HPG coating [2,23,30,31]". This reviewer finds no relation of what the authors say with reference [30]. For its part, reference [23] reviews the application of several imaging fusion techniques in the diagnosis and treatment of tumors, but does not mention anything about the decrease in saturation magnetization. At this point, this reviewer does not continue with the review for finding too many inconsistencies and abuse of self-citation, see [16], [24], [27], [28].

Not to mention that the manuscript requires a couple of English and structural revisions. All subscripts of chemical formulae in the references have been omitted. "Super Paramagnetic" should not be written separately.

Author Response

  1. Nanoparticles labeled with Sc and Sc for theranostic application” presents an idea of using superparamagnetic iron oxide-based nanoparticles (SPIONs) labeled with two scandium radionuclides, Sc and Sc, for visualization of cancer tissue, simultaneous direct irradiation, and local magnetic hyperthermia of the tumor, although the magnetization results do not offer this possibility. Consequently, to avoid confusion to the reader, the hyperthermia part should be omitted in the approach. It is striking that the authors measure the diameter in a cube. This is a misinterpretation of basic geometry. The diameter is the line segment that passes through the center and joins two opposite points of a circle. In 3D (sphere) it is defined as the segment that passes through the center and has its ends on the surface of the sphere. The authors should clarify this point. "...the Fe O NPs have a uniform cubic shape and a diameter from 38 to 50 nm"yapıldı.

You're right. As we wrote, because the magnetization of our nanoparticles was low, it is not possible to use hyperthermia process. We have removed relevant fragments from the text.

We changed diameter for “size”.

  1. There is too much combination of ideas that make the reader lose understanding. The authors should improve the writing of the manuscript. For example, in the first paragraph of the introduction the authors present several ideas that somehow try to link them, but this reviewer finds the approach confusing. It is preferable that the authors cite the original papers when they indicate that "micro and nanoparticles labeled Re, Au, In, Y, I and I have been prepared and used [3]." However, reference [3] is a paper where only Au is used and the others are listed in the introduction "Currently, many radionuclides are under intensive investigation for the therapeutic applications, particularly the Auger emitters..." this reviewer finds it bad practice to self-cite without clear justification.
    The authors indicate that the coating occurs before synthesis of the MNPs "MNPs can be coated with a biocompatible polymer before or after synthesis..." I do not find this statement meaningful.                                  We agree with the reviewer's comments. The introduction has been corrected. We deleted the first paragraph, We added also more references for all radionuclides.

  1. This reviewer does not find it valid to use a reference from almost 10 years ago to indicate that HPG has attracted attention in recent years. The authors indicate that "Shape and size, biocompatibility, biodegradability, and stability of nanoparticles are parameters to be considered." Were all of these parameters considered in their work, i.e., are there results for all of these?
    We rearranged the sentence ‘Among others, MNPs modified with Hyperbranched Polyglycerol (HPG) have attracted much attention for years’ and we added updated references. In addition, parameters like shape, size, biocompatibility, and stability of nanoparticles were studied. We deleted only biodegradability which was not performed.
  1. There are some statements without citing. As an example "The silica layer stabilizes the MNP core by sustaining magnetic dipole interactions." The authors should reinforce the ideas raised in the introduction by providing more references. For example, they could include https://doi.org/10.1016/B978-0-12-824364-0.00019-8 when talking about polymer-coated magnetic nanoparticles, they could also include https://doi.org/10.1155/2014/978284 to reinforce the approach of biomarker immobilization on silica-modified SPION surface.

We added two references according to your comments.

  1. The Objective of the work is clearly stated, however, it should not be written in the future. The final product is too complex, perhaps difficult to replicate even for the same research group, it is cubic iron oxide nanoparticles, coated with silica, then another component is added, HPG, later all this is conjugated with DPAPA, and even more with PSMA. Is it so? None of the results show evidence of the formation of the proposed material, and the interaction between the materials is not discussed. There is no evidence or demonstration that the nanoparticles have been modified. The authors describe a number of characterizations, such as FTIR and magnetization measurement without providing graphs. Why are the graphs not presented?

We added FTIR spectra (Figure 4) to our manuscript and analysed it.

  1. Figures 2C and 2D do not show any nanostructure, at least it cannot be seen at that SEM resolution. The authors indicate that "Because in our studies magnetite NPs were coated with nonmagnetic silica and HPG polymer, a similar mechanism could be considered to decrease saturation magnetization after silica and HPG coating [2,23,30,31]". This reviewer finds no relation of what the authors say with reference [30]. For its part, reference [23] reviews the application of several imaging fusion techniques in the diagnosis and treatment of tumors, but does not mention anything about the decrease in saturation magnetization. At this point, this reviewer does not continue with the review for finding too many inconsistencies and abuse of self-citation, see [16], [24], [27], [28]. Not to mention that the manuscript requires a couple of English and structural revisions. All subscripts of chemical formulae in the references have been omitted. "Super Paramagnetic" should not be written separately.

Thank you for your kind evaluation and comments. We have revised our manuscript according to your comments. Self-citation has been significantly reduced.

Reviewer 3 Report

The submitted manuscript reported development of multimodal radiobioconjugates of magnetic nanoparticles for diagnosis and treatment of prostate cancer. This topic is of interest for readers of Pharmaceutics. However, I have some reservations about the novelty and data provided. I therefore recommend publication of this manuscript only if the authors can address the major issues noted below.

1. The introduction section needs to be improved. The authors should highlight the significance clearly.

2. The figures are not of good quality, while the legend on the figure is not visible. All the figures need to be improved.

3. Some physicochemical properties of the nanoparticles are not well characterized. Dispersity is not provided.

4. The nanoparticle cellular uptake is not studied. It is necessary to carry out this study to understand the cellular interaction.

5. Groups should be correctly included and compared. In the figures, samples are not labelled properly for different groups. It is very difficult to compare the data.

6. Statistical analysis should be carried out to show if the results are significantly different.

7. 3D spheroids are not well defined and can’t see the details of nanoparticle distribution. Also lack of proper controls in that study. The authors need to properly design this study and provide more data.

8.  The writing of the manuscript needs be improved. Please also correct some typos and small errors.

Author Response

  1. The introduction section needs to be improved. The authors should highlight the significance clearly.

We agree with the reviewer's comments. The introduction has been significantly improved. We deleted also the first paragraph.

  1. The figures are not of good quality, while the legend on the figure is not visible. All the figures need to be improved.

We added high quality figures regarding to your comments.

  1. Some physicochemical properties of the nanoparticles are not well characterized. Dispersity is not provided.

We added the Polydispersity Index (PDI) of nanoparticles to the manuscript and analyzed it.

  1. The nanoparticle cellular uptake is not studied. It is necessary to carry out this study to understand the cellular interaction.

Cellular interaction was primarily investigated with receptor binding affinity studies as presented in manuscript. Due to characteristics of analyzed compound it is very difficult to investigate the cellular uptake of nanoconjugate. After internalization, each part of compound can be separately metabolized thus its proper evaluation could result in intricate layout of data. That is why we limited our research to most important and major for radiopharmaceuticals receptor binding affinity studies.

  1. Groups should be correctly included and compared. In the figures, samples are not labelled properly for different groups. It is very difficult to compare the data.

We labelled them properly. Thank you.

  1. Statistical analysis should be carried out to show if the results are significantly different.

Due to type of graph we chose this was impossible to implement statistical interpretation on figures 4 and 6 by asterix or other symbols. Instead of this we inserted into the manuscript text p-values to show statistical significance of obtained results discussed in the relevant section.

  1. 3D spheroids are not well defined and can’t see the details of nanoparticle distribution. Also lack of proper controls in that study. The authors need to properly design this study and provide more data.

Thank you for your kind evaluation and comments. We have revised our manuscript according to your comments.

  1. The writing of the manuscript needs be improved. Please also correct some typos and small errors.

We improved our manuscript regarding to your comments.

Round 2

Reviewer 2 Report

The authors addressed most of the recommendations or comments derived from the review. However, it must be said that they did not provide an answer to everything, for example they did not offer a response to the comment: The authors indicate that the coating occurs before synthesis of the MNPs "MNPs can be coated with a biocompatible polymer before or after synthesis..."

It should be noted that in the revised version there are still inconsistencies, for example, in line 423, where this reviewer recommended checking the information provided with the references indicated, it was left without references and with a text in Polish (according to DeepL) that should be clarified.

One of the main observations of this reviewer consisted of self-citation without being fully justified, the authors state that "Self-citation has been significantly reduced."

Author Response

We agree that the sentence "In medical applications, MNPs can be coated with a biocompatible polymer before or after synthesis to increase stability and bioavailability" The sentence was changed to "In medical applications, MNPs can be coated with a biocompatible polymer to increase stability and bioavailability.

We added also appropriate references in line 423. Polish text was removed. 

Reviewer 3 Report

The authors have addressed my comments and I am happy to support it to be published. 

Author Response

Thank you for accepting the revised version of the manuscript.